# Diurnal oscillations of MRI metrics in the brains of male participants

Matthew Carlucci [1,2], Tristram Lett[3], Sofia Chavez[4,5], Alexandra Malinowski[1], Nancy J. Lobaugh [4,6,7] & Art Petronis [1,2,5,7] ✉

Regulation of biological processes according to a 24-hr rhythm is essential for the normal functioning of an organism. Temporal variation in brain MRI data has often been attributed to circadian or diurnal oscillations; however, it is not clear if such oscillations exist. Here, we provide evidence that diurnal oscillations indeed govern multiple MRI metrics. We recorded cerebral blood flow, diffusion-tensor metrics, T1 relaxation, and cortical structural features every three hours over a 24-hr period in each of 16 adult male controls and eight adult male participants with bipolar disorder. Diurnal oscillations are detected in numerous MRI metrics at the whole-brain level, and regionally. Rhythmicity parameters in the participants with bipolar disorder are similar to the controls for most metrics, except for a larger phase variation in cerebral blood flow. The ubiquitous nature of diurnal oscillations has broad implications for neuroimaging studies and furthers our understanding of the dynamic nature of the human brain.

Cell-autonomous circadian oscillators and environmental cues such as light, sleep-wake, and feeding, interact across various animal tissues to produce diurnal rhythmicity[1]. Circadian and diurnal oscillations are an integral part of most biological processes including gene expression, metabolism, hormonal regulation, immune response, sleep, cognition, and behaviour[2,3]. Molecular oscillatory patterns are particularly complex in the brain, where different regions exhibit substantial variation in amplitudes and phases of oscillating RNA transcripts[4,5], metabolites[6], and proteins[7]. Yet, it is not clear if these periodic molecular effects translate into the larger-scale structural and functional features in the living human brain, which can be measured by magnetic resonance imaging (MRI).

In PubMed, we identified over 500 articles matching search parameters: 'diurnal OR circadian AND MRI AND brain'. In many of these publications, scanning the same participants twice a day detected morning-evening differences in metabolism/cerebral blood flow[8,9] brain/parenchymal volume[10,11], diffusion metrics[12–15], and parameters

derived from functional MRI[16,17]. These differences were not attributable to known MRI confounders such as technical differences in scanner type, image acquisition protocols, and data processing pipelines, nor to subject-related factors[10,11]. Therefore, it has often been assumed that these time-of-day effects reflect circadian or diurnal rhythmicity, but, in fact, their presence and parameters remain poorly understood. Thus far, the most direct experimental evidence for circadian oscillations in human MRI data comes from a functional MRI study that acquired several scans throughout the morning and night to test for changes in the brain's response to an attention task in conjunction with sleep deprivation[18]. To the best of our knowledge, no MRI studies to date have performed structural or quantitative imaging using an optimal design (sampling at evenly spaced intervals around the clock) to estimate 24-hr oscillations in the human brain.

The primary aim of the present investigation was to determine if 24-hr oscillations were present in an array of MRI metrics representing structural brain features and cerebral blood flow (CBF). As many of the

[1]The Krembil Family Epigenetics Laboratory, The Campbell Family Mental Health Research Institute, Centre for Addiction and Mental Health, Toronto M5T 1R8 ON, Canada. [2]Institute of Biotechnology, Life Sciences Center, Vilnius University, Vilnius LT-10257, Lithuania. [3]Center for Population Neuroscience and Precision Medicine (PONS), Clinic for Psychiatry and Psychotherapy, Charité Universitätsmedizin Berlin, Berlin 10117, Germany. [4]Brain Health Imaging Centre, Centre for Addiction and Mental Health, Toronto, ON, Canada. [5]Department of Psychiatry, Temerty Faculty of Medicine, University of Toronto, Toronto, ON, Canada. [6]Department of Medicine, Division of Neurology, Temerty Faculty of Medicine, University of Toronto, Toronto, ON, Canada. [7]These authors contributed equally: Nancy J. Lobaugh, Art Petronis. ✉e-mail: Art.Petronis@camh.ca

published studies mentioning diurnal or circadian rhythmicity examined CBF, diffusion, or metrics relying on T1 relaxation, we focused on related metrics for this work. Separation of endogenous (circadian) effects from those induced by environmental changes during day and night is not a trivial task. Therefore, we aimed to characterise diurnal rather than circadian rhythmicity.

Our secondary aim was to investigate if and how the oscillatory parameters differed in individuals affected by a brain disease. We recently suggested that disease may alter synchronous epigenomic circadian/diurnal oscillations[19,20], and this principle of desynchronosis is applicable to any other biological oscillator. Therefore, we additionally aimed to test the desynchronosis hypothesis in brain MRI metrics of persons diagnosed with bipolar disorder (BPD), a common psychiatric disorder with numerous changes in circadian regulation[21].

In this study (Fig. 1), we assessed 24 male participants (25–50 years of age, 16 controls and eight participants with BPD) each scanned every 3 hr over a 24-hr period, with the first scan beginning between 8:00 and 9:30. We derived nine MRI metrics from both standard and specialized MRI protocols (Methods) including arterial spin labelling, diffusion-tensor imaging, T1-weighted imaging, and T1 relaxometry. Each 30-min MRI session produced metrics of: cortical thickness (CT), cortical grey matter volume (GMV), cortical surface area (SA), white matter fractional anisotropy (WM-FA), mean diffusivity of white and grey matter (WM-MD, GM-MD), grey matter CBF, and white and grey matter quantitative longitudinal relaxation time (WM-qT1, GM-qT1).

Cosinor regression sinusoidal curve fits[22] (Methods) were applied to the brain-derived metrics, at the whole brain and regional level, to detect diurnal oscillations and estimate their parameters. We examined the MRI metric variation explained by the group-level oscillations and identified instances where oscillations were not synchronous across subjects. The analyses were extended to compare the BPD group with the controls.

## Results
### Participants
Twenty-four participants completed the study (Supplementary Fig. 1). Our inclusion criteria included both sexes; however, all participants self-identified as males. Six participants self-identifying as females expressed initial interest, and three were eligible, but due to scheduling challenges and the onset of the COVID-19 lockdown of the research facility, none participated. Six of the eight recruited BPD participants were being treated with a mood stabilizer (lithium, carbamazepine, lamotrigine, or valproic acid) at the time of scanning. One BPD participant was medication-free, but had a history of treatment with valproic acid, and one BPD participant's medication information was not disclosed. All BPD participants were in remission at the time of recruitment and during their scanning session. Their euthymic states were confirmed using the YMRS[23] -30 days prior to enrolment (mean[SD] = 30[16] days, range = 11–58 days). Apart from one Asian in the control group, the ethnic background of all participants was Caucasian. To ensure a streamlined experience for the BPD participants,

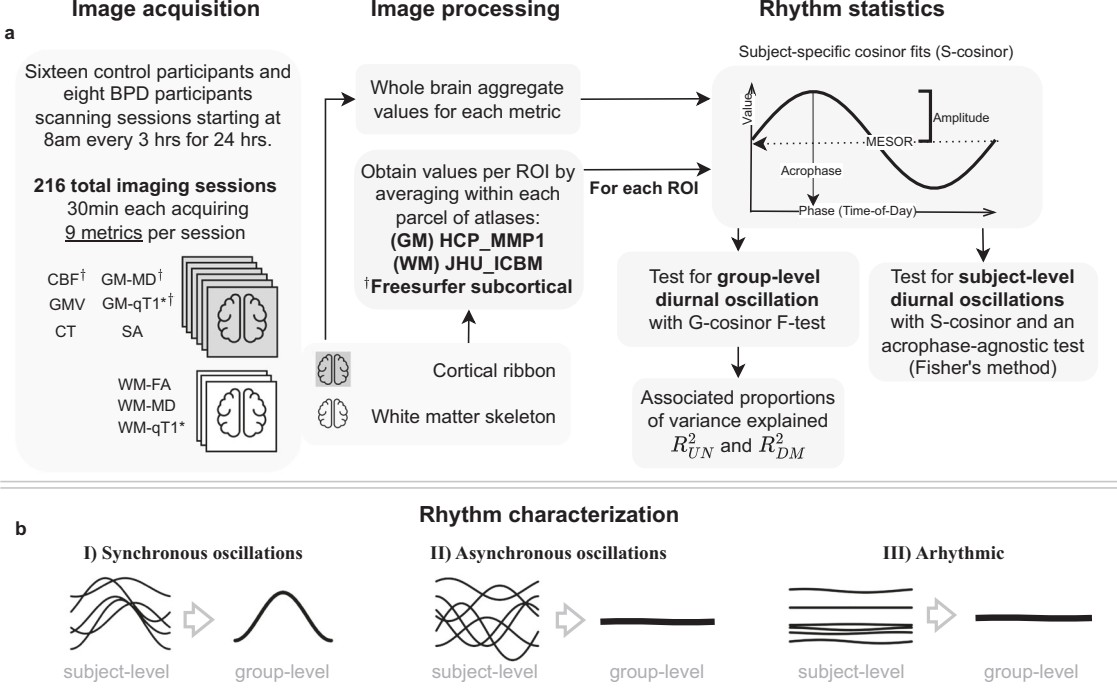

**Fig. 1 | Pipeline for estimating diurnal oscillations in MRI metrics. a** Image acquisition (left): Scanning schedule and metrics derived from imaging protocols. Grey brains: grey matter metrics (cortical ribbon, subcortical structures); white brains: metrics from white matter skeleton. Image processing (centre): Following registration to template, whole-brain aggregate values were obtained for the whole cortical ribbon and white matter skeleton (Supplementary Fig. 7) as well as parcels based on grey and white matter brain atlases. Rhythm statistics (right): Top: for each individual and metric, cosinor modelling produced estimates of the acrophase, amplitude, and MESOR. Bottom: G-cosinor test and an acrophase-agnostic test characterised group-level and subject-level oscillations, respectively and estimated proportion of variance explained by oscillations. **b** Three rhythmicity scenarios: (I) oscillations were present and synchronous across subjects, (II) oscillations were present but asynchronous across subjects, and (III) oscillations

were absent. Asterisk: qT1 was acquired for all participants with BPD, and 12 of the 16 control subjects. Dagger: Additional freesurfer automatic segmentation subcortical ROIs were obtained for these metrics. **a, b** Generated using app.diagrams.net v21.7.4. BPD participants with bipolar disorder, CBF cerebral blood flow, GMV grey matter volume, CT cortical thickness, GM-MD grey matter mean diffusivity, GM-qT1 grey matter quantitative T1 relaxation time, SA surface area, WM-FA white matter skeleton fractional anisotropy, WM-MD white matter skeleton mean diffusivity, WM-qT1 white matter skeleton quantitative T1 relaxation time, GM grey matter, HCP_MMP1 Human connectome project multi-modal parcellation atlas, WM white matter, JHU_ICBM John's Hopkins University, International Consortium of Brain Mapping atlas, MESOR midline estimating statistic of rhythm, $R^2_{UN}$ unnormalized proportion of variance explained, $R^2_{DM}$ normalized (demeaned) proportion of variance explained, ROI region of interest.

**Table 1 | Participant characteristics**

| Characteristic | Controls | BPD | t | p |
|---|---|---|---|---|
| Age | 35 (9.4) | 35 (4.7) | 0.056 | 0.96 |
| Weight [kg] | 85 (13) | 86 (9.8) | 0.25 | 0.80 |
| Height [cm] | 180 (5.5) | 170 (4.7) | −3.2 | 0.0053 |
| BMI | 26 (3.6) | 30 (3.3) 1 | 2.3 | 0.044 |
| Consumed caffeine | 12/16 | 8/8 | — | 0.26 |
| Is smoker | 1/16 | 4/8 | — | 0.028 |
| Actigraphy acro-phase [hr] | 16 (1.5) | 16 (2.3) | 0.056 | 0.96 |
| Actigraphy amplitude | 1.2 (0.35) | 1.1 (0.45) | −0.86 | 0.41 |
| Actigraphy MESOR | 2.6 (0.25) | 2.5 (0.24) | −0.79 | 0.45 |
| Actigraphy time-in-bed [hr] | 8.0 (1.7) | 7.3 (1.5) | −0.85 | 0.41 |
| Sleep duration [hr/day] | 7.1 (0.67) 4 | 7.2 (1.1) | 0.38 | 0.71 |
| Sleep quality [PSQI] | 5.1 (2.6) | 6.5 (3.4) | 1.0 | 0.33 |
| Exercise [hr/week] | 5.7 (4.9) 4 | 6.4 (9.8) | 0.20 | 0.85 |
| Body weight acro-phase [hr] | 24 (2.5) | 25 (2.0) | 1.0 | 0.32 |
| Body weight diurnal amplitude [kg] | 0.65 (0.29) | 0.65 (0.48) | 0.029 | 0.98 |
| Body weight MESOR [kg] | 86 (13) | 88 (13) | 0.41 | 0.69 |
| Young Mania Rating Scale | — | 1.0 (1.5) | — | — |

Continuous data are summarized as 'mean (SD) #missing' and were tested for group-wise dif-ferences between control (*n* = 16) and BPD participants (*n* = 8) with a two-sided Welch's two sample *t* test, while binary outcomes were tested with a two-sided Fisher's exact test with uncorrected p-values reported. Consumed caffeine: the participant consumed some caffeine (coffee, tea, soda) at various times and amounts during the on-site portion of the study. Is smoker: participant was a smoker/vaper; none consumed nicotine during the 24 hr scanning session. Exercise and sleep duration were self-reported. Weight diurnal oscillation parameters were estimated for each subject with a S-cosinor model (Methods). Actigraphy-based subject-specific diurnal parameters were estimated from log transformed actigraphy step data and average recorded total daily time-in-bed was obtained directly (Supplementary Fig. 5; Methods). *BPD* participants with bipolar disorder, *MESOR* midline estimating statistic of rhythm, *PSQI* Pittsburgh sleep quality index.

the first four scanning sessions were controls only, the next four sessions included participants with BPD (Methods). Control and BPD participants were similar in terms of most characteristics, including their Pittsburgh Sleep Quality Index[24] (PSQI) and sleep schedules in the week prior to scanning (Table 1).

## Technical variation
CBF or diffusion scans were repeated back-to-back three times at each of the nine time points in two control subjects per measure. Temporal variation across the 24-hr period was consistently significantly larger than within time point technical (scan-to-scan) variation for the relevant whole-brain metrics (CBF, GM-MD, WM-MD, and WM-FA; one-way ANOVA range $F_{8,18}[p]$ = 5.8[$9.4 \times 10^{-4}$] to 22[$7.8 \times 10^{-8}$]; Supplementary Table 1).

## Diurnal oscillations in the whole brain averages of control subjects
Whole-brain diurnal oscillations were tested with a two-stage cosinor model[22] (Methods) with a 24-hr period (Fig. 1a). Briefly, a subject-level cosinor regression model (referred to as S-cosinor going forward) was fit to each control subject's data to obtain subject-specific parameter estimates for: the midline estimating statistic of rhythm (MESOR), the oscillation amplitude, and the time of its peak (acrophase). These individual fits are shown in Fig. 2a. Then, an average group-level cosinor model (referred to as G-cosinor going forward) was obtained by averaging S-cosinor parameters[22] (Methods). The corresponding G-cosinor zero-amplitude F-test revealed that four of the nine imaging-

derived MRI metrics exhibited significant group-level 24-hr oscillations (Supplementary Table 2). In order of acrophase time these were: WM-MD ($F_{2,14}$ = 10.5, $p$ = 0.0017, acrophase = 13 hr), GM-MD ($F_{2,14}$ = 9.5, $p$ = 0.0025, acrophase = 16 hr), WM-FA ($F_{2,14}$ = 8.8, $p$ = 0.0034, acro-phase = 18 hr), and CBF ($F_{2,14}$ = 6.8, $p$ = 0.0088, acrophase = 21 hr). Individual acrophases and corresponding significant group-level acrophases with their 95% confidence intervals (CIs) are shown in Fig. 2b, and a summary of diurnal oscillation amplitudes and other related statistics for each significant MRI metric are presented in Table 2.

The proportion of variance explained ($R^2$) by these diurnal oscillations was assessed under two contexts. First, we proceeded without adjusting for inter-individual variation in subjects' MESORs to obtain an unnormalized $R^2$ ($R^2_{UN}$). Here, we utilized the G-cosinor MESOR, amplitude, and acrophase to model the data. The CBF G-cosinor model explained the most total variance ($R^2_{UN}$ = 0.020), followed by WM-MD ($R^2_{UN}$ = 0.0098), GM-MD ($R^2_{UN}$ = 0.0055), and WM-FA ($R^2_{UN}$ = 0.0034) (Fig. 2c). Second, the proportion of variance explained by diurnal oscillations was assessed after controlling for inter-individual variation by demeaning the data within each subject. This effectively set the MESOR to zero, and we utilized only the amplitude and acrophase estimates of the G-cosinor model. The proportion of remaining variance in the demeaned data (demeaned $R^2$; $R^2_{DM}$) explained by diurnal oscillations increased by about an order of magnitude compared to $R^2_{UN}$, with the largest in GM-MD ($R^2_{DM}$ = 0.20), followed by WM-MD ($R^2_{DM}$ = 0.19), CBF ($R^2_{DM}$ = 0.15), and WM-FA ($R^2_{DM}$ = 0.084) (Fig. 2d).

No significant group-level diurnal oscillations were found for CT, GMV, SA, GM- and WM-qT1, which suggested that these metrics were static, or oscillations were too small to be detected with the current sample size (this is further explored in the BPD section). Alternatively, within-subject diurnal oscillations may have been real, but out of phase across individuals, i.e., asynchronous oscillations. In this case, aggregation of widely dispersed individual acrophases would have led to a G-cosinor amplitude too low to produce significant group-level oscillations (Fig. 1b, II). To test this alternative possibility, we utilized the S-cosinor p-values and Fisher's meta-analytic method[25] to estimate the accrued evidence for subject-level diurnal oscillations (Methods). This acrophase-agnostic test was significant for both GM-qT1 ($p$ = 0.0042) and WM-qT1 ($p$ = 0.042), but not for CT, GMV, and SA (Supplementary Table 2).

To provide context to the observed oscillations in the brain with non-brain metrics known to oscillate diurnally, we also measured subjects' body weights prior to each scan session. Weight exhibited strong evidence for 24-hr oscillations with its acrophase around mid-night (G-cosinor $F_{2,14}$ = 32, $p$ = $5.7 \times 10^{-6}$, acrophase = 24 hr, $R^2_{UN}$ = 0.00041, $R^2_{DM}$ = 0.38, Table 2, Supplementary Fig. 2), which was very similar to previously published results[26]. Body weight co-varies with total body water[26], and direct experimental manipulation of water intake results in changes to brain volumes[27,28] and quantitative MRI metrics[29]. Based on such findings, body weight has been used as a proxy for hydration status[14]. In our study, including subject weight as a covariate did not eliminate oscillatory effects; all four MRI metrics remained significant (all G-cosinor $p$ < 0.05; Supplementary Table 3).

## Diurnal oscillations in brain regions of control subjects
We partitioned the brain into predefined regions of interest (ROIs) to test whether diurnal oscillation characterisation remained feasible at the regional-level (see Methods for details). Briefly, CBF, GM-MD, GM-qT1, SA, CT, and GMV values were obtained from 358 cortical grey matter ROIs using the Human Connectome Project Multi-Modal Parcellation cortical atlas[30] (HCP_MMP1), an additional 14 subcortical ROIs from the Freesurfer automatic segmentation atlas[31] were obtained for CBF, GM-MD, and GM-qT1; WM-MD, WM-FA, and WM-qT1 values were obtained from 46 skeletonized white matter ROIs using the Johns Hopkins University diffusion-based white-matter atlases[32].

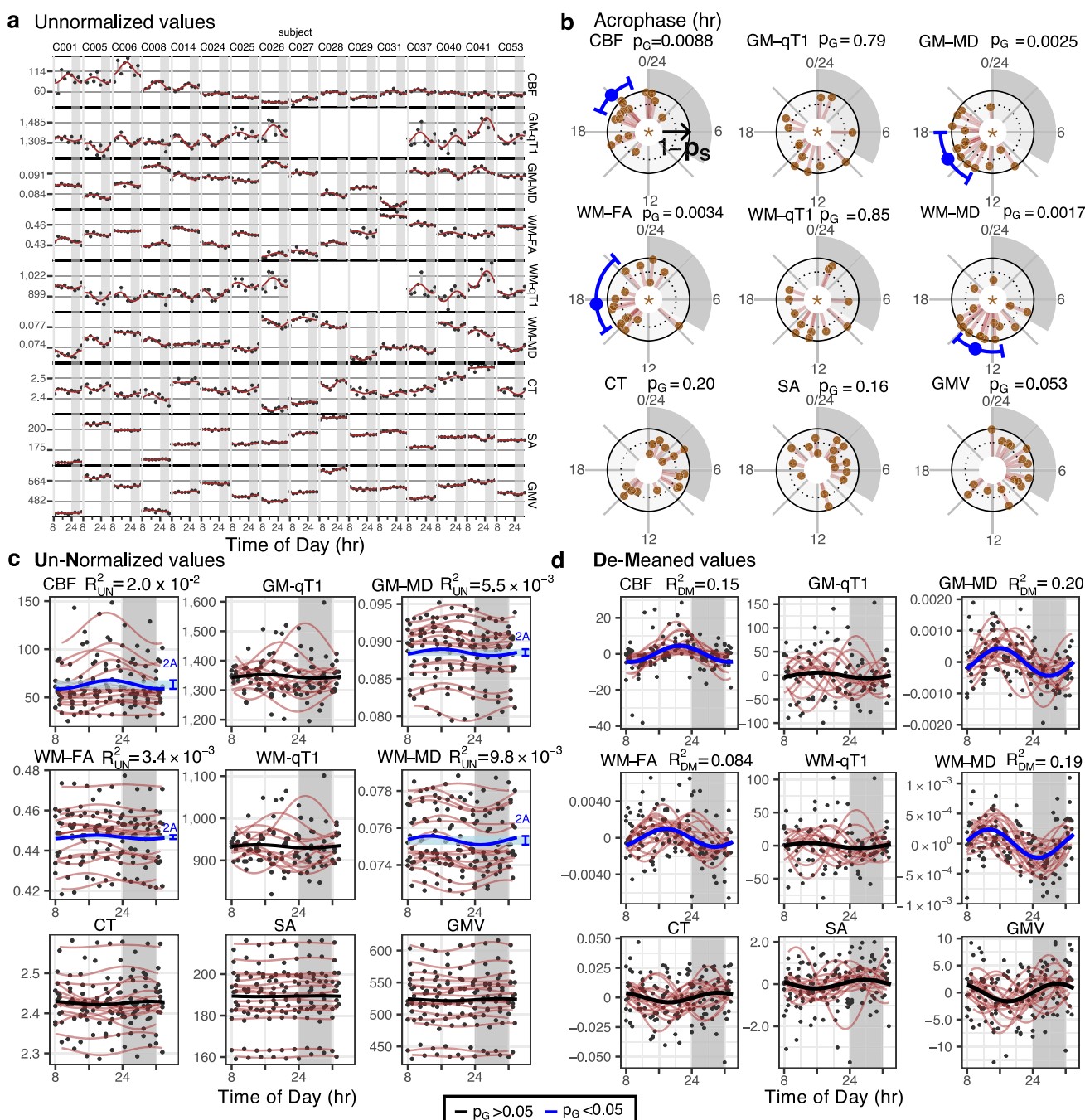

**Fig. 2 | Diurnal oscillations in the whole brain. a** Metric values (y-axis) over a 24-hr period (x-axis) for each subject (columns; $n = 16$) for each of the nine MRI metrics (rows). S-cosinor fit lines are shown with residuals connecting dots to the fit. **b** Circular plots of S- and G-cosinor acrophases. Rotational axis: time of day (0-24 hr). Brown dots: S-cosinor acrophase for each subject. Radial axis: inverse S-cosinor $p$ value from a one-sided $F$ test ($1-p_S$) ranging from zero to one. Blue lines: significant ($p_G < 0.05$) G-cosinor acrophase estimates with corresponding 95% confidence interval. Plot panel titles indicate the G-cosinor p-values from a one-sided $F$ test ($p_G$). Panel centres: Star indicates metrics where the acrophase-agnostic test was significant (one-sided chi-square test, $p < 0.05$). **c** Raw metric values as in (**a**), grouped by metric, with S-cosinor fit lines in brown; G-cosinor curve shown in blue where the blue bracket and shading indicate peak-to-peak amplitude (2*A) of

the G-cosinor model. Plot panel titles indicate the proportion of variance explained by the G-cosinor model ($R^2_{UN}$). **d** Within-subject demeaned values (black dots) reveal oscillatory effects of the G-cosinor model, represented as thick lines (black, $p_G > 0.05$; blue, $p_G < 0.05$). Brown lines: Individual S-cosinor fits. Plot panel titles indicate the proportion of variance explained by the G-cosinor model on the demeaned data ($R^2_{DM}$). **a**–**d** qT1 was not acquired for four subjects. **b**–**d** All p-values are not corrected for multiple testing. See Supplementary Table 2 for statistics. CBF cerebral blood flow, GM-qT1 grey matter quantitative T1 relaxation time, GM-MD grey matter mean diffusivity, WM-FA white matter skeleton fractional anisotropy, WM-qT1 white matter skeleton quantitative T1 relaxation time, WM-MD white matter skeleton mean diffusivity, CT cortical thickness, SA cortical surface area, GMV cortical grey matter volume. Source data are provided as a Source Data file.

We found significant evidence of regional diurnal oscillations using identical methods to whole-brain analysis, for each ROI. S-cosinor regression was first applied for every subject, MRI metric, and ROI, independently (Fig. 3a; Supplementary Fig. 3). Resulting

group-level oscillations were nominally significant (G-cosinor $p < 0.05$) in 5.9% to 85.8% of ROIs across seven of the nine MRI metrics (Supplementary Table 4). After false-discovery rate (FDR) adjustment, we obtained significant ($q < 0.05$) regional diurnal oscillations for CBF,

**Table 2 | MRI metrics showing significant G-cosinor results**

| | | Whole brain | | | | | | |
| | | Oscillation parameters | | | G-cosinor zero amplitude | Effect size | | |
| | Metric | MESOR [CI] | Amplitude [CI] | Acrophase [CI] | $p$ | $R^2_{UN}$ | $R^2_{DM}$ | Units |
|---|---|---|---|---|---|---|---|---|
| Controls (n = 16) | CBF | 63 [51,76] | 4.5 [2.0,7.0] | 21 [20,23] | 0.0088 | 0.020 | 0.15 | ml g$^{-1}$ min$^{-1}$ |
| | GM-MD | 0.089 [0.087,0.091] | 4.4e-04 [2.3e-04,6.5e-04] | 16 [14,18] | 0.0025 | 0.0055 | 0.20 | 0.1 mm$^2$ s$^{-1}$ |
| | WM-FA | 0.45 [0.44,0.45] | 9.9e-04 [3.8e-04,0.0016] | 18 [16,21] | 0.0034 | 0.0034 | 0.084 | — |
| | WM-MD | 0.075 [0.074,0.076] | 2.4e-04 [1.3e-04,3.5e-04] | 13 [11,15] | 0.0017 | 0.0098 | 0.19 | 0.1 mm$^2$ s$^{-1}$ |
| | Weight | 86 [79,93] | 0.55 [0.38,0.72] | 24 [1,23] | 5.7e-06 | 4.1e-04 | 0.38 | kg |
| BPD (n = 8) | GM-MD | 0.092 [0.089,0.094] | 4.8e-04 [2e-04,7.6e-04] | 15 [12,19] | 0.027 | 0.0094 | 0.15 | 0.1 mm$^2$ s$^{-1}$ |
| | WM-FA | 0.45 [0.44,0.45] | 0.0019 [0.0012,0.0025] | 17 [15,19] | 0.0019 | 0.015 | 0.23 | — |
| | SA | 185 [171,198] | 0.52 [0.20,0.84] | 3 [1,7] | 0.030 | 8.1e-04 | 0.14 | 10$^3$ mm$^2$ |
| | Weight | 88 [77,99] | 0.61 [0.21,1.0] | 24 [2,23] | 0.034 | 0.0046 | 0.37 | kg |
| Combined (n = 24) | CBF | 59 [50,67] | 3.2 [1.3,5.2] | 21 [20,23] | 0.0090 | 0.013 | 0.10 | ml g$^{-1}$ min$^{-1}$ |
| | GM-MD | 0.090 [0.088,0.091] | 4.5e-04 [3e-04,6.1e-04] | 16 [14,17] | 2.9e-05 | 0.0049 | 0.18 | 0.1 mm$^2$ s$^{-1}$ |
| | WM-FA | 0.45 [0.44,0.45] | 0.0013 [8e-04,0.0018] | 17 [16,19] | 1.5e-05 | 0.0059 | 0.13 | — |
| | WM-MD | 0.075 [0.075,0.076] | 2.1e-04 [1.3e-04,2.9e-04] | 13 [12,15] | 0.00012 | 0.0084 | 0.18 | 0.1 mm$^2$ s$^{-1}$ |
| | SA | 188 [182,194] | 0.31 [0.13,0.49] | 4 [1,7] | 0.0053 | 1.7e-04 | 0.043 | 10$^3$ mm$^2$ |
| | GMV | 519 [500,537] | 1.3 [0.34,2.3] | 6 [3,9] | 0.038 | 5.4e-04 | 0.048 | 10$^3$ mm$^3$ |
| | Weight | 86 [81,92] | 0.57 [0.42,0.72] | 24 [1,23] | 6.1e-07 | 0.0018 | 0.38 | kg |

| | | Regional ranges for FDR q < 0.05 ROIs | | | | | | |
| | Metric | MESOR | Amplitude | Acrophase | % p < 0.05 (#; # q < 0.05) | $R^2_{UN}$ | $R^2_{DM}$ | Units |
|---|---|---|---|---|---|---|---|---|
| Controls | CBF | 46–82 | 1.8–9.2 | 18–23 | %85.8 (319/372; 289) | 0.0062–0.047 | 0.033–0.21 | ml g$^{-1}$ min$^{-1}$ |
| | GM-MD | 0.073–0.10 | 4.5e-04–0.0013 | 14–20 | %37.9 (141/372; 53) | 5.95e-04–0.030 | 0.052–0.21 | 10$^3$ mm$^2$ |
| | WM-MD | 0.071–0.085 | 2.1e-04–0.0013 | 8–18 | %26.1 (12/46; 3) | 0.0042–0.028 | 0.096–0.13 | 10$^3$ mm$^2$ |
| BPD | WM-FA | 0.51–0.67 | 0.0028-0.0046 | 15–20 | %21.7 (10/46; 5) | 0.006–0.035 | 0.084–0.25 | — |
| | CT | 2.5 | 0.062 | 4 | %3.6 (13/358; 1) | 0.096 | 0.19 | mm |
| Combined | CBF | 42–72 | 1.7–6.1 | 18–24 | %83.3 (310/372; 265) | 0.0045–0.034 | 0.028–0.18 | ml g$^{-1}$ min$^{-1}$ |
| | GM-MD | 0.072–0.11 | 2.7e-04–0.0015 | 8–19 | %55.4 (206/372; 168) | 0–0.024 | 0.024–0.18 | 10$^3$ mm$^2$ |
| | WM-FA | 0.49–0.66 | 2.1e-03-0.0029 | 13–18 | %34.8 (16/46; 8) | 0.0038–0.013 | 0.046–0.11 | — |

Group-level cosinor (G-cosinor) parameters and statistics are presented for controls only, BPD participants only, and the combined group. Supplementary Table 2 and Supplementary Table 4 provide statistics and summaries for all metrics. Reported $p$ values are not corrected for multiple testing.

*MESOR* midline estimating statistic of rhythm, *CI* confidence interval, *CBF* cerebral blood flow, *GM-MD* grey matter mean diffusivity, *WM-FA* white matter skeleton fractional anisotropy, *WM-MD* white matter skeleton mean diffusivity, *SA* cortical surface area, *GMV* cortical grey matter volume, *CT* cortical thickness, *BPD* participants with bipolar disorder, $R^2_{UN}$ unnormalized proportion of variance explained, $R^2_{DM}$ normalized (demeaned) proportion of variance explained, *FDR* false discovery rate, *ROI* region of interest.

GM-MD, and WM-MD (289, 53, and 3 ROIs, respectively; Fig. 3b, c). The acrophases of these ROIs ranged from 18–23 hr in CBF, 14–20 hr in GM-MD, and 8–18 hr in WM-MD. These regional acrophases were generally within the 95% confidence interval (CI) of their respective whole-brain acrophases; maximal ROI differences from the whole-brain acrophases were 3 hr for CBF (whole-brain CI spans 3 hr), 4 hr for GM-MD (CI spans 4 hr), and a 5 hr difference originating from two of the three oscillating WM-MD ROIs (CI spans 4 hr) (Fig. 3c; Table 2). $R^2_{UN}$ effect sizes were distributed around their observed whole brain $R^2_{UN}$ values, with ROIs reaching $R^2_{UN} = 0.047$, 0.030, and 0.028 for CBF, GM-MD and WM-MD, respectively. After within-subject demeaning, ROI $R^2_{DM}$ were lower than their observed whole brain $R^2_{DM}$ on average, yet maximum $R^2_{DM}$ were near whole brain observations at $R^2_{DM} = 0.21$, 0.21 and 0.13 for CBF, GM-MD and WM-MD, respectively (Table 2). These regional diurnal oscillation statistics are shown in Supplementary Fig. 4.

The remaining metrics did not exhibit group-level oscillations at the regional level. We observed that the regional acrophases for GM-qT1 (Fig. 3a) and WM-qT1 (Supplementary Fig. 3) were clustered within each subject, but these clusters were dissimilar across subjects. This suggested that individual diurnal oscillations were present regionally, but group-level regional oscillations were undetectable due to the highly variable acrophases across subjects. The follow-up acrophase-agnostic test detected 40.6% and 8.7% of ROIs with nominally significant evidence for subject-level diurnal fits in GM-qT1 and WM-qT1, respectively, with 4 ROIs surviving FDR adjustment ($q < 0.05$) in GM-qT1 (Supplementary Table 4).

### Diurnal oscillations in bipolar disorder
After establishing diurnal oscillations in controls, we turned our attention to the group of eight participants diagnosed with BPD. We

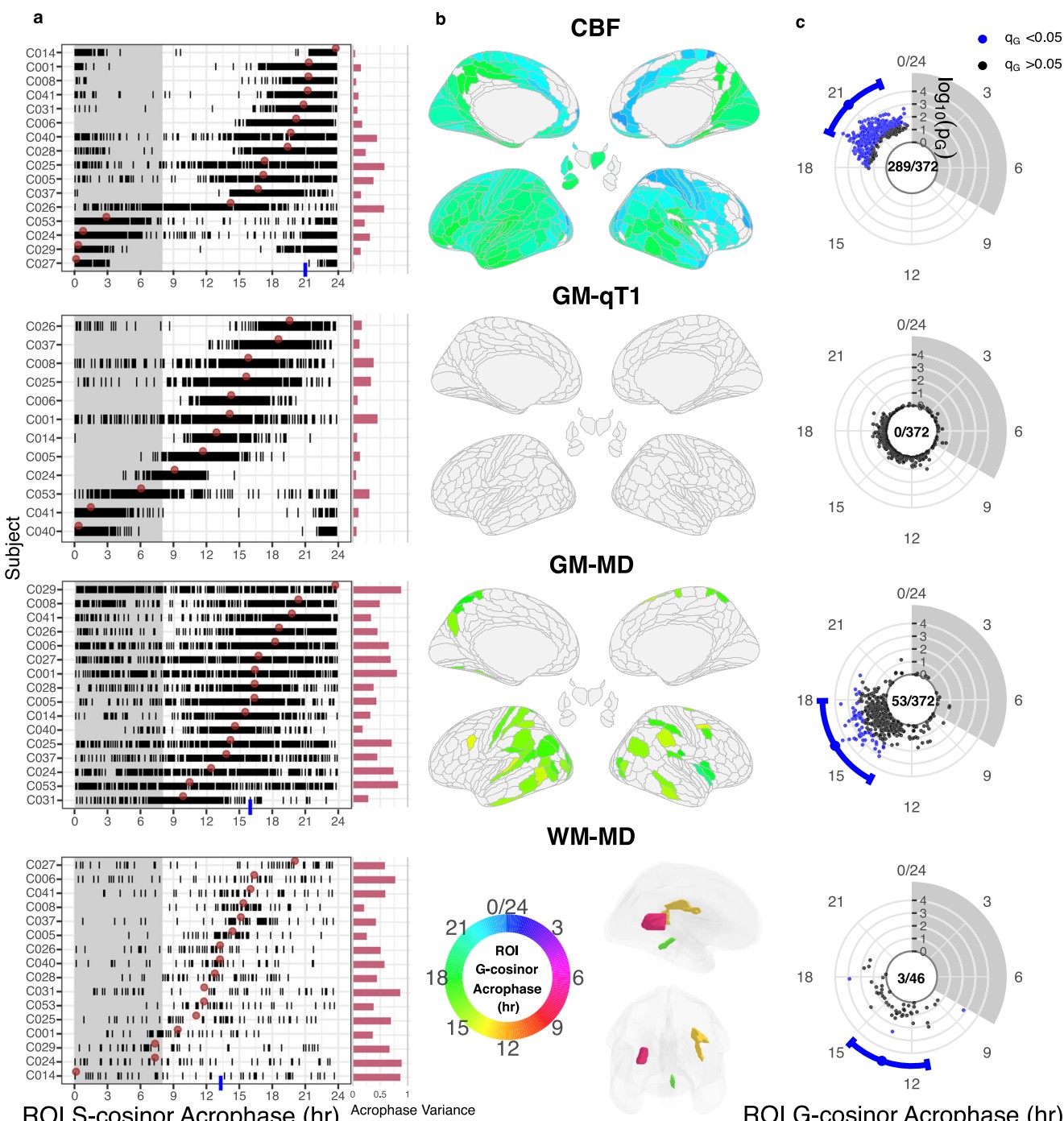

**Fig. 3 | Regional cosinor statistics for control subjects.** ROI-wise results for CBF, GM-MD, and WM-MD to illustrate significant group-level oscillations, and absence of such in GM-qT1. Each metric is shown in one row as indicated by centre titles. **a** Left: Subject-level cosinor (S-cosinor) acrophase estimates are plotted as vertical ticks for each ROI (x-axis), individually for each subject (y-axis). For each metric, subjects are sorted according to their whole brain subject-level acrophase (brown dots). Blue tick: significant ($p_G < 0.05$) whole-brain group-level cosinor (G-cosinor) acrophase estimate. Right: Red bars indicate each subject's acrophase variance across all ROIs (Methods). **b** Colour-coded G-cosinor acrophase times for significant ROIs (FDR $q_G < 0.05$) from a one-sided $F$ test displayed on brain surfaces.

Grey regions were not significant. **c** Blue dots: estimates of the G-cosinor acrophase for each ROI shown on the rotational axis in polar coordinates where the radial axis is the negative logarithm of the uncorrected G-cosinor $p$ value from a one-sided F-test ($p_G$). Blue intervals: significant whole brain acrophases and 95% CI. Panel centres: the number of significant ROIs to total ROIs. All 9 MRI metrics are presented in Supplementary Fig. 3. CBF cerebral blood flow, GM-qT1 grey matter quantitative T1 relaxation time, GM-MD grey matter mean diffusivity, WM-MD white matter skeleton mean diffusivity, ROI region of interest, FDR false discovery rate, CI confidence interval. Source data are provided as a Source Data file.

first established that neither their diurnal actigraphy or body weight oscillatory parameters differed from the controls (Table 1; Supplementary Fig. 5). Next, we interrogated the BPD MRI data for evidence of oscillations using the same analytical methods as for controls

(Fig. 4a), tested for BPD-control differences, and then combined the two datasets to increase the sample size for further oscillation detection (Fig. 4a–c). Finally, we followed up on evidence for desynchronosis in CBF and its related implications to disease studies (Fig. 4d, e).

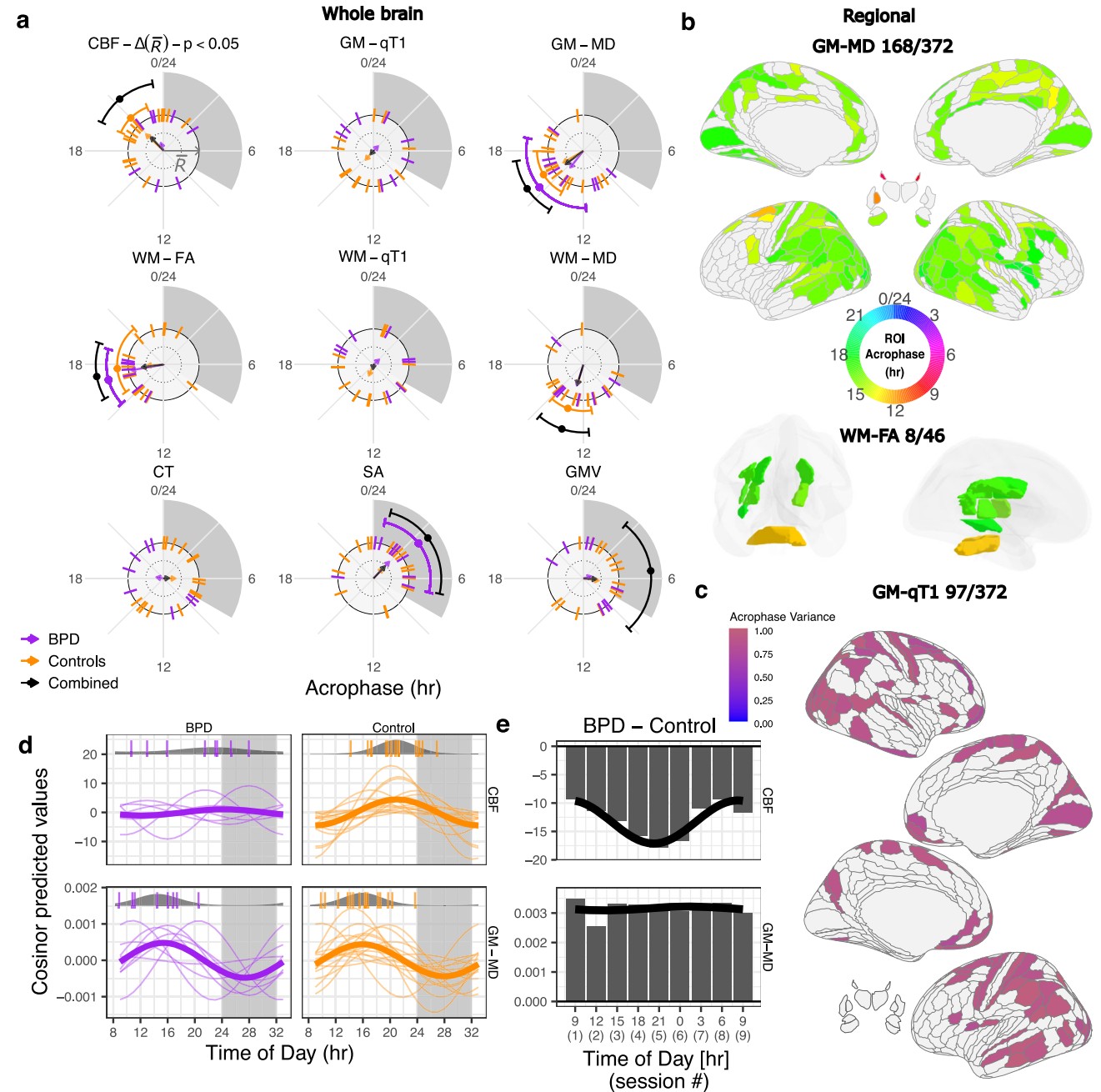

**Fig. 4 | Diurnal oscillations in BPD, control, and the combined group. a** Raleigh plot with acrophases of whole-brain oscillations (G-cosinor one-sided F-test $p < 0.05$) and 95% confidence intervals shown on an outer orbit. S-cosinor acrophases for each subject are shown as ticks. Each arrow points in the direction of the circular mean acrophase across control subjects (orange arrow), participants with BPD (purple), or both (black) and its length represents a measure of acrophase consistency across subjects (1 − acrophase variance). The significant difference (permuted $p < 0.05$) in acrophase consistency between BPD participants ($n = 8$) and controls ($n = 16$) for CBF is indicated in the panel title. $p$ values are not corrected for multiple testing. **b** Acrophases of regional oscillations (G-cosinor one-sided $F$ test FDR $q < 0.05$) for GM-MD and WM-FA in the combined group (as in Fig. 3b) illustrating the increased number of oscillating ROIs relative to controls-only. **c** Significant ROIs for GM-qT1 in the combined group (one-sided chi-square acrophase-agnostic test FDR $q < 0.05$) coloured according to their acrophase variance.

**d** S-cosinor models (thin lines) on the demeaned data in CBF and GM-MD with G-cosinor model (thick line). Top of each panel: S-cosinor acrophases shown as ticks overlaid on probability density plots (circular normal) estimated for each group. **e** Between-group differences in CBF and GM-MD means at each of the nine scanning sessions, highlighting the impact of the oscillation differences between BPD participants and controls (x-axis: average session time). A cosinor fit to the differences is shown as a black line. CBF cerebral blood flow, GM-qT1 grey matter quantitative T1 relaxation time, GM-MD grey matter mean diffusivity, WM-FA white matter skeleton fractional anisotropy, WM-qT1 white matter skeleton quantitative qT1 relaxation time, WM-MD white matter skeleton mean diffusivity, CT cortical thickness, SA cortical surface area, GMV cortical grey matter volume, ROI region of interest, FDR false discovery rate, BPD participants with bipolar disorder. Source data are provided as a Source Data file.

At the whole-brain level, the BPD group exhibited significant 24-hr rhythmicity for GM-MD and WM-FA (G-cosinor $F_{2,6} = 6.9$, $p = 0.027$, and $F_{2,6} = 21.3$, $p = 0.0019$, respectively; Fig. 4a). Acrophase times for the two metrics in BPD (15 hr and 17 hr, respectively) did not differ from the control group (16 hr and 18 hr, respectively; permutation $p = 0.77$, and $p = 0.68$, respectively; Methods). Taken together, these findings argue that oscillations were not altered in BPD for GM-MD and WM-FA.

Several MRI metrics did not show consistent evidence for group-level whole-brain oscillations in both groups (nonsignificant in BPD: CBF, WM-MD; nonsignificant in controls: SA; nonsignificant in either group: GMV, CT, WM- and GM-qT1). These metrics were not tested for any group-wise differences in their G-cosinor parameters. We assumed that there may have been inadequate power to consistently detect oscillations. If subthreshold oscillatory signals had similar acrophases, they could become significant when assessed in a combined cohort of all 24 subjects. All metrics that were significant in the G-cosinor test in one of the two cohorts were also significant when the data were combined (Fig. 4a; Supplementary Table 2). The same was true for WM- and GM-qT1 in the acrophase agnostic test. However, GMV, initially considered to be arhythmic in both groups, passed the threshold for significant oscillations in the combined sample ($F_{2,22} = 3.8$, $p = 0.038$; acrophase = 6 hr). The single remaining measurement arhythmic in both groups was CT, which did not show evidence of synchronous or asynchronous oscillations in the larger sample (Fig. 4a; Supplementary Table 2).

At the brain regional level, the BPD cohort alone produced only five significantly oscillating ROIs (G-cosinor FDR $q < 0.05$; all in WM-FA), likely due to the cohort's small sample size. Adding this cohort to the controls, however, increased the number of significant ROIs in many metrics compared to the controls only (Supplementary Table 4). G-cosinor detected an additional 117 and 8 ROIs for GM-MD and WM-FA (Fig. 4b), plus 17 additional ROIs for CBF (despite 24 fewer ROIs overall; Supplementary Table 4) in the combined sample. The acrophase-agnostic test revealed 93 additional ROIs (FDR $q < 0.05$) for GM-qT1 (Fig. 4c), plus 119, 101 and 2 additional ROIs for CBF, GM-MD and WM-MD, respectively (Supplementary Table 4). Altogether, these findings suggest that power is limited in both cohorts at the regional level.

Unlike the controls, CBF in the BPD participants did not show group-level oscillations at the whole-brain level (G-cosinor $p = 0.74$). Two explanations for the lack of significance of a CBF group-level oscillation in the BPD-only analysis were considered: (1) a loss of individual oscillations in the participants with BPD (Fig. 1b, III), or (2) a loss of acrophase consistency relative to controls, i.e., desynchronosis (Fig. 1b, II). Our analyses did not detect evidence for the first possibility; individual CBF oscillation strengths in the BPD participants did not differ from those in the controls (differential $R^2$ of the within-subject S-cosinor fits; two-sided Welch's $t$ test, $t(15) = 0.39$, $p = 0.70$), indicating no loss of individual oscillations in BPD. In support of the second possibility, the acrophases of the BPD participants were widely distributed around the clock (Fig. 4a). The BPD acrophase variance was significantly larger than the controls (acrophase variance = 0.77 and 0.32 in BPD participants and controls, respectively; permutation $p = 0.017$; Methods). Therefore, we concluded that desynchronosis contributed to the lack of a BPD group-level oscillation. Interestingly, the magnitude of each BPD participant's acrophase deviation from the combined cohort strongly correlated with their subjective sleep quality score (PSQI, Pearson's $r = 0.91$; two-sided $p = 0.0015$; Supplementary Fig. 6).

The absence of CBF group-level oscillations in BPD participants, but presence of such in the controls, translated into a disease-related time-of-day effect. To put this finding in the context of cross-sectional studies, we compared the means of the participants with BPD and controls at each time point and showed that the magnitude of CBF differences exhibited a significant oscillation (cosinor $F$ test $p = 0.0025$; Fig. 4d, e). The differences in whole-brain blood flow in the BPD group changed two-fold from 9 ml g$^{-1}$ min$^{-1}$ below controls at 8 hr to 18 ml g$^{-1}$ min$^{-1}$ below controls at 21 hr. In contrast, for metrics where oscillations did not differ between the two groups, group-wise differences were stable across the 24-hr period (e.g. GM-MD, cosinor $F$ test $p = 0.92$; Fig. 4d, e).

## Discussion

In this study, we have shown 24-hr oscillations in eight of nine human brain measurements. Many of these phenomena have been discussed frequently in the literature[8–17], but were not formally identified prior to this study. Two statistical approaches, G-cosinor modelling and the acrophase-agnostic test, identified two facets of oscillation: one where the oscillations were synchronous across subjects at the group level (G-cosinor), and one where oscillations were strong at the individual level (S-cosinor), but asynchronous across subjects. The two effects are not mutually exclusive, and most G-cosinor significant metrics were also significant in the acrophase-agnostic test. On the other hand, WM- and GM-qT1 exhibited strong subject-level diurnal effects detected only by the acrophase-agnostic test.

Diurnal oscillations in the brain exhibited several interesting features across the MRI metrics, individuals, and brain regions. In the whole brain, a non-negligible proportion of the total variance was explained by diurnal oscillations across MRI metrics ($R^2_{UN} = 0.34\%–2.0\%$). On the other hand, when inter-subject variability in steady-state values was removed, the model explanatory power improved, and diurnal oscillations were shown to be a substantial source of within-subject variation ($R^2_{DM} = 8.4\%–20\%$). All other factors being equivalent, data acquired in a typical cross-sectional neuroimaging study will reflect the combination of differences in the MESOR and the presence of oscillations. These effects may vary spatially (Supplementary Fig. 4), and certain brain regions may show stronger oscillations than others, however, a number of factors contribute to their detectability (e.g., sample size, effect size, ROI reliability, etc.). Considering oscillations such as these, and their regional variation, can reduce variability and biases in populational MRI studies. Strategies will depend on study size, design, and goals and may include: fixed time-of-day sampling, randomization, collecting time-of-day information, and modeling oscillatory effects.

Evidence for 24-hr oscillations provides insights into the common, but previously experimentally unproven, interpretation that time-of-day effects are fragments of circadian/diurnal variation e.g., diffusion metrics in WM[13]. Technical and methodological disparities prohibited direct comparisons of our findings with the existing time-of-day literature, yet our oscillatory models may help explain the origins and variability of morning-evening differences. Demonstrably, if an acrophase (or nadir) occurs around noon (e.g. WM-MD; acrophase = 13 hr) morning-evening differences may be sensitive to scanning time (Fig. 5, red arrows). Here, even small changes in the timing of morning and evening scans can lead to inconsistent results. Time-of-day effects in other MRI metrics may be less sensitive to scan times, e.g., in WM-FA (acrophase = 18 hr), where evening scans would consistently produce a stronger signal compared to morning scans (Fig. 5, black arrows).

Uncovering the mechanisms for the observed diurnal oscillations, as well as previous time-of-day effects[8–17], is part of a major and fundamental question: how are MRI measurements impacted by a variety of lower-level physiological and molecular brain processes. We made an initial step in this direction by determining that brain water content did not explain our results (Supplementary Table 3). The broad range of whole-brain acrophases across MRI metrics (Table 2) suggests other singular explanatory factors are implausible. Even data derived from a single pulse sequence (DTI) showed whole-brain acrophase variability from 13–18 hr (GM-MD:16 hr; WM-MD:13 hr; WM-FA:18 hr, Table 2). Multifaceted, multi-level approaches are necessary to reveal the mechanisms for MRI oscillations and to explain their individual, regional, and disease-related differences.

Our findings provide a foundation to guide designs (e.g., scan timing and sample sizes) for future studies of additional temporal patterns and oscillations in the brain. We demonstrated that, after adding the BPD group to the controls, additional oscillating ROIs were identified at the group level (GM-MD and WM-FA; Fig. 4b), and

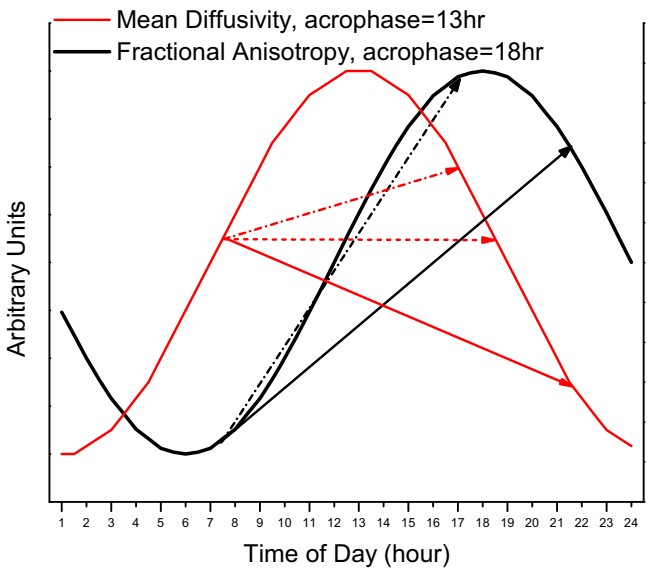

**Fig. 5 | Group-level cosinor models for fractional anisotropy and mean diffusivity in the white matter skeleton (controls only).** For mean diffusivity (red), acrophase was at mid-day (13 hr). For fractional anisotropy, (black), acrophase was in late afternoon (18 hr). Arrows indicate possible time-of-day effects. Solid arrows: Scans at 7:30 and 21:30; Dash-dotted arrows: Scans at 7:30 and 17:00; Dotted arrow (mean diffusivity only): Scans at 7:30 and 18:30. Acrophases from Table 2.

subject-level (CBF, GM-qT1 and GM-MD; Fig. 4c). ROI oscillation maps for most brain metrics, however, remained sparse, indicating that larger studies are necessary for identification of diurnal effects at regional and/or voxel-based resolution.

Subject- or population-specific diurnal oscillations of MRI-based brain features may help to uncover etiopathogenic mechanisms of neuropsychiatric disease. As exemplified by CBF, oscillation characteristics can differ significantly in persons diagnosed with BPD compared to controls. CBF desynchronosis suggests the presence of diurnal shifts in metabolic demand and adds to the numerous molecular[21] and behavioural[33] facets of circadian/diurnal dysregulation in BPD. A subset of BPD participants, however, exhibited CBF acrophases within the acrophase range of control subjects, which may indicate a stable remission. Alternatively, a large acrophase deviation from the norm may indicate poorly controlled remission or perhaps even a looming relapse of disease. The preliminary link between objective acrophase measures and the BPD-participant-reported sleep quality (Supplementary Fig. 6) may help uncover the mechanisms of disturbed sleep and hallmark BPD relapse[34].

There are some limitations of our study, some of which can be addressed in the future. First, due to restrictions during the COVID-19 lockdown era, we were not able to scan female subjects and the sample size of the group of participants with BPD was limited to eight. Second, we did not control for the impact of nutrition, exposure to light, seasonal variation, and other Zeitgebers, which could have differentially impacted the BPD and control cohorts. However, the observed similarities in diurnal MRI metrics between groups and across subjects argues against substantial externally induced biases. Third, the array of medications taken by the BPD participants may have affected their diurnal oscillations. Fourth, our study was not sufficiently powered to perform some comparisons, such as ROI oscillations, either across regions or across the two groups, both of which are of critical importance for uncovering the brain disease topography. Finally, waking up multiple times during the night for scanning disrupted sleep patterns and may have confounded MRI measurements. This limitation, however, is hard to address when studying the living human brain.

Mapping individual-specific rhythms may help address a major drawback of traditional cross-sectional studies, especially in clinical populations: there is abundant evidence that group-averaged measurements poorly represent individual disease states[35]. MRI desynchronization parameters may provide the basis from which to establish novel biomarkers, subsequently enabling the stratification of disease subtypes and explaining the contradicting results of chronotherapeutics[36]. Patient-specific brain-regional and/or temporal features of disease may become of particular importance in the personalization of targeted therapeutic approaches such as deep brain stimulation.

## Methods

### Data collection

**Participants.** Participants were considered for inclusion if they were between 18–50 years of age and met the inclusion criteria for their group. Exclusion criteria that applied to all participants were: pregnancy, blindness, metabolic disease (e.g., diabetes), dieting, and contraindications to MRI. All assessments were performed at the Centre for Addiction and Mental Health (CAMH) in Toronto, ON, Canada.

**Controls.** Individuals were recruited via word-of-mouth referrals, existing study registries and advertisements. Inclusion criteria were no history of alcohol or substance use disorders, psychiatric disorders, neurological disorders or sleep disorders. Smoking was not an exclusion criterion due to the frequent consumption of nicotine products in participants with BPD.

**Participants with BPD.** Participants with BPD were recruited from existing patient registries and databases at CAMH. The same inclusion/exclusion criteria used for controls applied to the BPD group, with the exception of a history of psychiatric disorder. To be included, a diagnosis of BPD-I or BPD-II was required, which was confirmed using the Mini-International Neuropsychiatric Interview[37]. The Young Mania Rating Scale (YMRS) was administered to confirm the BPD participants were euthymic.

The CAMH Research Ethics Board approved the study. All participants provided written informed consent prior to commencement of the study and in accordance with the Declaration of Helsinki. Participants were remunerated for each scanning session.

**General procedures.** During the week prior to the scanning sessions, participants wore a FitBit Flex™ for 4–5 days to achieve three full days of recordings (see *Actigraphy*). All participants completed the PSQI at the time consent was obtained. On the day of scanning, participants arrived at the CAMH imaging facility between 7:00 and 9:00, ~1 hr before their first scan. Participants changed into a surgical gown/pants and were provided private rooms, which included a desk and bed for the duration of the ~25 hr study. Each subject was scanned every 3 hr for a total of nine ~30-min MRI sessions; scheduled scan start time was assigned to the session. In between scanning sessions, participants had access to regular meals (9:30–10:30, 13:30–14:30, and 19:30–20:30), snacks, water and coffee, all of which were recorded, but not quantified, and were not controlled for time of day. Participants were permitted to go about their regular activities while on site (including work, leisure, eating and sleeping).

During the preparation phase (September 2017–August 2018), we decided which MRI metrics should be interrogated, optimised the scan frequency, which initially was set to be every 6 hr but after preliminary analysis, was changed to every 3 hr, and resolved various logistical issues related to the ~25 hr experiment. All scan sessions were performed over 8 weekends, from Saturday morning to Sunday morning, spread over a 15-month period from November 2018 to February 2020. Control subjects were scanned from November 2018 to August 2019. Recruitment of BPD participants was launched in January 2019, after

evidence of diurnal oscillations for several MRI metrics was detected in the controls. Without such evidence, collection of data from participants with BPD would have not been justified. Therefore, BPD participants were scanned from August 2019 to February 2020.

**Actigraphy.** The FitBit data were acquired primarily to confirm self-reported sleep durations and PSQI (reported in Table 1); we did not attempt to assign a chronotype to each person. Upon consent into the study, participants were provided with a Fitbit Flex 2 (firmware 24.24.30.2) and were requested to wear the device for 4-5 full days prior to the study date. Upon receipt of the device, participants were guided to download the official Fitbit smartphone app from the corresponding (Android or iOS) app store, and were instructed to synchronize their device data every 6 hr with the corresponding account. A final synchronization was performed on the morning of the study at which time the devices were returned. Device data were extracted from the official Fitbit website using the Request Data export function (February 2020), and were obtained in JSON format. Sleep-wake cycles and by-minute step data were included in the raw exported data. Step data were extracted from the raw download step JSON files where 72 hr of contiguous data (3 days starting from midnight) were obtained for each subject. Data were imported as Greenwich Mean Time and converted to their respective Toronto time. One participant's actigraphy data were collected for the 5 days following the study visit due to synchronization issues.

**Image acquisition.** MRI data were acquired using a 3.0-Tesla GE Discovery MR750 (General Electric Medical Systems, Milwaukee, WI, USA, DV26.0_R02_1810.b). Each MRI session included the following acquisitions: T1-weighted imaging (3D BRAVO, sagittal slices, 0.9 mm³ voxels, echo time: 3.02 ms, repetition time: 6.77 ms, flip angle: 8°), 3D pseudo-continuous arterial spin labelling imaging (pCASL, 3.0 mm³ voxels, axial slices, echo time: 11.11 ms, repetition time: 5050 ms, flip angle: 111°, post-label delay: 2025 ms), diffusion-weighted imaging ($b$ value = 1000, 2 mm³ voxels, 32 diffusion directions; posterior to anterior encoding direction; four $b$ value = 0, repetition time ~7142 ms). An 8-volume non-diffusion weighted sequence was also acquired with the same parameters, but with the encoding direction anterior to posterior for B0-induced distortion correction. Four acquisitions were used to calculate calibrated quantitative T1 relaxation time maps with B1 correction[38] (sagittal slices, two high resolution (1mm³) fast spoiled gradient echo scans with whole-brain excitation, echo times: 4.4 ms; repetition times 10.6 ms; and flip angles of 3°(Flip3) and 14° (Flip14); two lower resolution (4 mm³) SPGR scans with repetition times 50–60 ms, echo time 5 ms, flip angles 130° and 150°).

**Technical variation.** To measure variance due to scanner performance (technical variation), DTI and ASL measurements were each repeated three times (15–18 min of acquisition time) at each of the nine timepoints. On the same scanning day, DTI and ASL were repeated for two separate pairs of subjects (Diffusion: C027 and C029; CBF: C028 and C031). For those subjects, the quantitative T1-mapping protocol was dropped to accommodate the extra scans, and their second DTI or ASL scans were used for all other analyses.

## Image processing
**T1-weighted.** T1-weighted image processing was accomplished using Freesurfer (version 7.1.1, http://surfer.nmr.mgh.harvard.edu/fswiki) and Advanced Normalisation Tools (ANTs) version 2.3.3 (http://stnava.github.io/ANTs)[39]. Following N4 bias correction[40], as implemented in ANTs, unbiased (equidistant from all sources) within-subject T1-weighted templates were created using all T1-weighted images for each participant (nine images per subject) to create 24 individualised templates with the Freesurfer mri_robust_template script[41]. Each

participant's T1-weighted image was linearly registered to their respective template image using ANTs. For each registered T1-weighted image, brain extraction was performed using the antsBrainExtraction.sh script and the OASIS-30 Atropos template[42] (https://mindboggle.info/data.html). Cortical reconstruction was performed on all T1-weighted images using the Freesurfer image analysis suite. After Freesurfer's recon-all step1, the ANTs brain extracted mask was applied to Freesurfer's watershed brain mask to provide a tighter brain extraction for surface creation. Each brain extraction was manually checked, and the final steps of recon-all were performed.

**Diffusion.** Diffusion-weighted images (DWI) were processed using FSL (version 6.0.3, https://fsl.fmrib.ox.ac.uk/fsl/fslwiki/FSL). Eddy current correction and echo-planar B0-induced imaging distortion correction were performed using the phase-encode reversed non-DWIs ($b = 0$) with topup[43] and eddy[44]. FSL's dtifit was used to calculate fractional anisotropy (FA) and mean diffusivity (MD). For registration purposes, unwarped average non-DWIs were calculated, and within-subject unbiased non-DWI templates were created using the mri_robust_template. Subject-specific tract-based spatial statistics (TBSS, https://fsl.fmrib.ox.ac.uk/fsl/fslwiki/TBSS)[45] skeletons were created using a modified version of TBSS (build e934eb2, https://github.com/trislett/ants_tbss) that implements ANTs linear and non-linear transformations prior to skeletonization instead of FSL's FLIRT and FNIRT. For the mean FA image, a FA cutoff of 0.2 was applied to create the mean FA skeleton. The unbiased, within-subject non-DWI template also underwent linear and non-linear transformations to the subjects' N4-corrected T1-weighted image to transform the MD images into native T1-weighted space. Technical issues were identified for C063, session 2, so all of their MRI metrics were reduced to eight time points.

**Cerebral blood flow.** The default CBF maps calculated by the scanner software were used for analysis. The difference images (untagged-tagged) generated by the scanner software first underwent linear and non-linear registration to the brain-extracted non-DWI ($b = 0$) template for each subject. The CBF maps were then transformed to non-DWI template space, and the CBF maps were transformed to the unbiased T1-weighted volume using the same transformation used to move the subject's MD images to T1-weighted space.

**Quantitative T1 mapping.** All four SPGR images used to create the qT1 maps were first reoriented and transformed into the space defined as halfway between the Flip3 and Flip14 images using the halfway_flirt command from FSL's SIENA pipeline (https://fsl.fmrib.ox.ac.uk/fsl/fslwiki/SIENA). The transformed Flip3 image was brain-extracted using FSL's Brain Extraction Tool (bet, https://fsl.fmrib.ox.ac.uk/fsl/fslwiki/BET) and the resulting mask was used to extract the brain of the reoriented Flip14 image. B1 maps were generated using the two high flip angle scans via the method of slopes[46], and qT1 maps were computed using the variable flip angle method with a B1 correction[46] and calibration procedure[38]. Linear and non-linear transformations of the qT1 maps to the unbiased T1-weighted subject images were performed and the qT1 maps were transformed to template space using the T1-weighted unbiased subject to MNI152 1 mm space linear and non-linear transformations. qT1 data quality was deemed to be not usable due to participant motion for one subject (C056) on their ninth scan, so their qT1 data were reduced to eight time points.

## Regions of interest
**Grey matter ROIs.** The Human Connectome Project Multi-Modal Parcellation atlas[30] (HCP_MMP1, MMP1.0 210 V, https://balsa.wustl.edu/976l8) was applied to the Freesurfer cortical tessellation to generate 358 cortical regions of interest (ROIs) to extract regional CT, surface area (SA), and grey matter volume (GMV). To generate mean

regional CBF, GM-MD, and GM-qT1 values, ROIs from the HCP_MMP1 atlas were transferred into each subject's T1-weighted space using multiAltasTT (V0.0.1, https://github.com/faskowit/multiAtlasTT) incorporating FreeSurfer gaussian classifier surface atlases[47]. For GMV, GM-MD, CBF, and GM-qT1, 14 additional subcortical regions were obtained using Freesurfer's automatic segmentation (aseg) of bilateral thalamus, caudate, putamen, amygdala, pallidum, and accumbens[31]. Bilateral hippocampus was included in both the HCP_MMP1 and the Freesurfer atlases; the Freesurfer parcellation was used. Mean CT, mean GMV, and total SA from the Freesurfer vertices were used to generate the measures for whole-brain analyses. A whole-brain cortical grey matter mask was derived from all HCP_MMP1 atlas regions and was used to calculate mean grey matter values for CBF, GM-MD, and GM-qT1. The whole-brain mean grey matter measures do not include the Freesurfer subcortical parcels.

**White matter ROIs.** The parcellations from the Johns Hopkins University DTI-based white-matter ICBM atlas[48,49] as provided in the FSL software (JHU ICBM DTI-81; JHU_ICBM) were used to define 46 white matter ROIs on the TBSS skeleton and to extract regional and whole-skeleton mean white matter MD (WM-MD), FA (WM-FA), and WM-qT1 (Supplementary Fig. 7). Note that bilateral tapetum was not included due to missing data in some subjects. For all metrics, data were obtained from the TBSS skeleton and ROIs in native T1-weighted space. For consistency with the description of the grey matter whole-brain results, whole-skeleton measures are referred to as 'whole-brain' in the main text.

**Units.** Unless otherwise indicated, metrics are in units as follows: CBF, ml g$^{-1}$ min$^{-1}$; GMV and subcortical volume, $10^3$ mm$^3$; SA, $10^3$ mm$^2$; CT, mm; FA, unitless [1]; MD, 0.1 mm$^2$s$^{-1}$; qT1, ms.

## Data analysis and statistical methods

**Subject-level oscillations (S-cosinor).** For each subject, cosinor linear regression was fitted with a 24-hr period to detect diurnal oscillations[22]. The data were modelled by linear cosine and sine transformations of MRI acquisition time ($t$) to arrive at a model for MESOR, amplitude, and acrophase, starting with the following formula:

$$y = \beta_0 + \beta_1 \cos\left(\frac{2\pi t}{\tau}\right) + \beta_2 \sin\left(\frac{2\pi t}{\tau}\right) + \text{error} \quad (1)$$

where $t$ is MRI acquisition time, $\tau$ is the period (24-hr). The intercept coefficient ($\beta_0$) is the MESOR. $\beta_1$ and $\beta_2$ are regression coefficients that were used to calculate the amplitude ($A$) and acrophase as:

$$A = \sqrt{\beta_1^2 + \beta_2^2} \quad (2)$$

$$\text{acrophase} = \text{atan2}(\beta_1, \beta_2)\frac{\tau}{2\pi} \quad (3)$$

Such that $y = \beta_0 + A\cos(\frac{t-acrophase}{2\pi\tau}) + \text{error}$. Each subject's p-values for the significance of diurnal oscillations were determined by the $F$ test comparing the cosinor model to an intercept-only (null) model and were used for the acrophase-agnostic test (described below). $R^2$ of these within-subject models were obtained for each subject and used for the CBF differential oscillation strength test.

**Group-level oscillations (G-cosinor).** The population-mean cosinor[22] approach was applied with the aim to make inferences regarding a populational average rhythm for each group, and is referred to as G-cosinor throughout the text. This is a two stage model where first stage estimates were obtained by fitting each subjects' data to the S-cosinor model described above. Next, across subjects, the

coefficients were averaged to obtain: $\beta_0^*$ (MESOR), $\beta_1^*$ and $\beta_2^*$, and the following group-level model:

$$\hat{y} = \beta_0^* + A^*\cos(t - \text{acrophase}^*) \quad (4)$$

Where '*' indicates a population mean estimate, and $A^*$ and acrophase$^*$ were derived from $\beta_1^*$ and $\beta_2^*$. In this procedure, S-cosinor [$\beta_1$,$\beta_2$] values were considered jointly. A single hypothesis test was performed with the null that the true population mean values are zero (i.e., [$\beta_1^*$,$\beta_2^*$] = [0, 0]) or equivalently, that $A^* = 0$. The p-value was obtained by the corresponding multivariate $F$ test[22] and is referred to as G-cosinor p-value throughout the text. In the regional analyses, for each metric, correction for multiple comparisons was performed using the Benjamini-Hochberg FDR[50] procedure. 95% CIs for $A^*$, acrophase$^*$, and $\beta_0^*$ were estimated by population-mean cosinor methods[51] deriving from the multivariate $F$ test.

**Variance explained.** A proportion of variance explained was calculated as $R^2_{UN} = 1$-SSR/TSS where the G-cosinor model was used to obtain:

$$\text{SSR} = (y - \hat{y})^2 \quad (5)$$

$$\text{TSS} = (y - \beta_0^*)^2 \quad (6)$$

$R^2_{DM}$ was calculated identically to $R^2_{UN}$ except y was first mean subtracted within each individual's data, effectively setting each subject's $\beta_0$ to zero and controlling for interindividual variation in $\beta_0$. Therefore $\beta_0^*$ of the G-cosinor models was also set to zero and we proceeded to calculate $R^2_{DM}$. One oscillating ROI in the GM-MD combined analysis had a low magnitude negative $R^2_{UN}$ estimate which was set to zero for clarity.

**Differential acrophase.** When BPD and control groups each had significant G-cosinor oscillations, a difference in their acrophases was tested with permutation testing. Acrophase$^*$ was obtained for both BPD (acrophase$^*_{BPD}$) and control (acrophase$^*_{CONTROL}$) groups. The minimum difference between them (minimum circular arc length) was then calculated. A permuted null distribution was generated by obtaining the same acrophase difference 10,000 times after shuffling BPD/control labels among subjects and recalculating group-wise acrophase$^*$ estimates. The two-sided permutation $p$ value was the proportion of instances where an absolute acrophase difference from a permuted trial was larger than the real difference.

**Acrophase-agnostic test for asynchronous subject-level diurnal oscillations.** We implemented a method to complement the G-cosinor test to better utilize the variance explained by each individual within-subject S-cosinor fit. The S-cosinor F-test provided significance for each subject's fitted cosinor curve. As such, individual tests were not influenced by the amplitude/acrophase of other subjects, and were evaluated only by the degree to which they explained the within-subject data. We considered the global null hypothesis that no subjects showed oscillations. Under this null, p-values should follow a uniform distribution, and Fisher's combined probability test[25] considers a likely distribution of p-values (as they would be if the global null were true) with the statistic:

$$\chi_{2k}^2 \sim -2\sum_{i=1}^{k}\log(p_i) \quad (7)$$

Where it follows a chi-square distribution with 2k (k = number of subjects) degrees of freedom. The test is conditional on the independence of subjects and is unweighted with respect to within-subject sample size.

**Acrophase variance.** Utilizing standard circular statistics[52], with a list of acrophases ($\theta$ in radians), an estimate of their mean and variance was obtained by representing each of them as a point, $u(C,S)$, on the unit circle as:

$$u(C, S) = (\cos(\theta), \sin(\theta)) \qquad (8)$$

The mean (centroid) $\bar{u}$ of these points was obtained as:

$$\bar{u}(\bar{C}, \bar{S}) = \left( \frac{1}{k} \sum_{i=1}^{k} \cos(\theta_i), \frac{1}{k} \sum_{i=1}^{k} \sin(\theta_i) \right) \qquad (9)$$

Then, the length of $\bar{u}$ is a measure of acrophase consistency:

$$\bar{R} = \sqrt{\bar{C}^2 + \bar{S}^2} \qquad (10)$$

where 1 indicates all acrophases were identical, and conversely, circular variance is $1 - \bar{R}$ and is bounded from 0–1. The mean acrophase was obtained from the direction of $\bar{u}$ as $\bar{\theta} = \mathrm{atan2}(\bar{C}, \bar{S})$.

**Differential acrophase variance.** Acrophase variance for both BPD ($1 - \bar{R}_{\mathrm{BPD}}$) and control ($1 - \bar{R}_{\mathrm{CONTROL}}$) groups were calculated as above. The difference between them (BPD − controls) was calculated. A permuted null distribution was generated by obtaining the same acrophase variance difference 10,000 times after shuffling BPD and control labels among subjects and recalculating group-wise acrophase variance estimates. The one-sided permuted p-value was the proportion of instances where the permuted acrophase variance difference was greater than the real difference.

**Plotting conventions.** Approximate dark times are shown as midnight to 8:00 for all relevant visualizations.

**Implementation and visualizations.** Statistical analysis was performed in R ver. 4.1.3. The Scikick command-line tool[53] was used to execute and archive all data analysis results. R package ggplot2 v3.4.0[54] was used for generating figures and package ggseg v1.6.6[55] was used to produce all brain images.

### Reporting summary
Further information on research design is available in the Nature Portfolio Reporting Summary linked to this article.

## Data availability
All processed data generated in this study are available as a dataset[56] on the Zenodo platform (https://doi.org/10.5281/zenodo.8360149). This dataset contains subject-level: processed whole-brain and ROI MRI data, body-weight data, and processed actigraphy data. Anonymized subject-level data are available to other investigators under restricted access in compliance with institutional ethics and privacy policies. Access requests can be submitted via Zenodo. The raw MRI data are protected and are not available due to institutional ethics and privacy restrictions. Source data are provided with this paper.

## Code availability
The code used to generate statistical results and figures is available as a software archive[57] on the Zenodo platform (https://doi.org/10.5281/zenodo.8360408; https://github.com/matthewcarlucci/DiurnalMRI).

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

## Acknowledgements

We thank the MRI technologists: Anusha Ravichandran, Hillary Bruce, Garry Detzler, and Valerie MacDonald for data collection; CAMH Epigenetics lab staff: Akhil Nair, Miki Susic, Michael Sherman, Richie Jeremian, Edward Oh, Aiping Zhang, William Che, Sasha Ebrahimi, Jeff Villamil, Turner Silverthorne for assistance with various parts of the project; Anthony Randal McIntosh for his comments on the manuscript. This project received funding from European Social Fund (project No 09.3.3-LMT-K-712-17-0008) under grant agreement with the Research Council of Lithuania (LMTLT) to A.P. The project was also supported by grants from Brain Canada and the CAMH Foundation, Project #554, from the Canadian Institutes of Health Research: TGH-158223 and PJT-148719, and the Krembil Foundation (CAMH Foundation Project #186) to A.P.

## Author contributions

A.P. conceived the idea of testing diurnal rhythmicity in the brain. N.J.L., S.C. and A.P. designed the overall study protocol and MRI acquisition details. S.C. calculated the qT1 maps. T.L. was responsible for the image processing. M.C., T.L. and N.J.L. conducted the data analyses. A.M. was responsible for the study logistics and conducted in-depth literature reviews. All authors participated in the writing and editing of the manuscript.

## Competing interests

The authors declare no competing interests.
