## [Peer Review File · Nature Communications]

Diurnal oscillations of MRI metrics in human brainsReviewer #1 (Remarks to the Author):

This manuscript describes the results of a study that measured functional and structural MRI characteristics repeatedly (every three hours over 24 hours) in men with (n=16) and without (n=8) bipolar disorder (BD). Evidence for diurnal oscillations were observed in healthy individuals in white and gray matter mean diffusivity, white matter fractional anisotropy, and cerebral blood flow (CBF) and in gray and white matter quantitative relaxation time for meta-analytic analyses that were agnostic to acrophase; results held after controlling for significant body weight oscillations. Similar oscillations were observed regionally, particularly for CBF. BD showed similar oscillations for gray matter mean diffusivity and white matter fractional anisotropy, but not CBF or white matter mean diffusivity. BD showed larger phase variation in brain perfusion and the degree of deviation from the group mean was correlated with subjective sleep quality. Depending on the time of day, there were two-fold differences between BD and controls in CBF. Combining the samples revealed evidence for diurnal oscillations in surface area and gray matter volume not seen in controls or BD alone. The authors interpret their findings to have implications for population MRI studies and identify the need for higher power studies to examine, in particular, regional oscillations. They also discuss that desynchronization of CBF might be a biomarker of risk for relapse or a way to understand sleep disturbance in the disorder.

The study is novel and important and the manuscript is well written and clear. Although the sample sizes are small, particularly for the BD group, the within-subject design was powered well for the global measures. Only men were studied, which is acknowledged as a limitation for generalizability.

The methods are appropriate to the questions posed. I was a bit unclear about whether, in the regional analyses, the global effects were somehow controlled or regressed out in order to identify regionally-specific rhythms? Also, were the regional measures weighted at all by their relative size (and presumably, reliability)?

The conclusions follow logically from the findings. My main critique is that the authors did not discuss possible biological or environmental bases for the observed diurnal oscillations. What mechanisms might lead to the particular shape of the rhythms seen for, say, CBF in controls? Particularly for the more structural measures, what's going on in the brain that leads to different diffusivity at different times of the day? In addition, it might help researchers in the field for the authors to make some concrete recommendations about how to best "consider" these oscillations (especially the ones that are asynchronous across individuals) in cross-sectional designs given that most studies will not be able to scan participants 9 times in one day.

Reviewer #2 (Remarks to the Author):

The authors present a very interesting study on a circadian oscillation in different MRI measures in the brain. Scanning protocol including sessions every 3h for 24h is a novel approach that was not introduced in previous neuroimaging research. The work will undoubtedly be of great importance in the field of chronobiology and neuroimaging. I only have a few remarks.

The analysis was performed on a relatively small group. Have sample size calculations been performed?

According to General Procedure description, all subjects started scanning protocol at the same time (approx. 9 am). Did the authors consider the randomization of starting sessions across 24h?

The authors declared to share processed data. Would you consider uploading raw (only

anonymized) data to an open repository? This would be of great benefit and enable further analysis of this valuable data set.

In the Discussion section, there is a lack of results description in relation to the literature, as well as a discussion of the results in the context of circadian changes in the physiology of the brain.

Reviewer #3 (Remarks to the Author):

In this article, the authors investigated with the help of repeated MRI acquisitions diurnal oscillations in a group of healthy adults and a smaller group of bipolar disorder patients.

The article is well-written and focuses on an important aspect which is broadly overlooked or neglected in many neuroimaging studies. Further, the authors used an advanced analysis method to detect possible oscillatory effects in various MRI-based measures. However, the study got abruptly stopped because of the COVID-19 pandemic. Although the study is well-designed and well-performed, this is a substantial limitation of the study, which propagates through all analyses and the validity of the conclusions.

My main concern is the combination of the two groups. There are several issues with that procedure. Of course, it is unfortunate that the BDP group is so small, but it is still reasonable to expect at least some deviations.

First of all, I strongly recommend not combining the two groups and then arguing whether significant results improved or not and using this as a proxy for whether BDP groups differed in this parameter or not. Given that there are potential differences, like in the CBF measure, indicates that there are potential differences between the groups, as predicted. The absence of clear effects in the BDP group for other parameters does not indicate that there are no effects. Lumping together the two groups on this weak evidence will primarily increase the chance of generating a statistical artefact rather than a valid result.

Secondly, the authors might also consider skipping the BDP group for this report and only reporting on the control group. This would make the study much more stringent and easier to interpret. I also encourage the author to continue their BDP study as planned. With the results provided now, it is difficult to draw any meaningful conclusions for this particular group.

Thirdly, I'm missing an in-depth discussion on the causality of the observed effects. Why do the observed changes occur? Could it be that the observed effects in GM-MD and WM-MD are indirect effects of variations in CBF? I'm thinking here along the line of partial volume effects in the diffusion data caused by CBF (and perhaps related CBV) effects.

Similarly, the introduction should also be revised to give a clearer picture of why these measures have been selected and what the expectations are.

Other comments:

- Line 202 & 203: How were the results interrogated for evidence of oscillations? Only by eyeballing? And how was the similarity assessed?
- Line 354 & line 377: There is conflicting information on how long the participants wear the actigraphs
- Line 361: Did the authors collect information on how much coffee, tea, or nicotine the participants consumed between scanning?
- Similarly, did the author collect data for blood pressure?

I'm looking forward to seeing a revised version of this manuscript

Karsten Specht

RESPONSE TO THE REVIEWERS' COMMENTS

All reviewers, in one way or another, asked for further discussion of the causality underlying the diurnal MRI findings. Our study was designed to uncover *whether* rather than *why* oscillations are present. A determination of causality would require more extensive biophysical modeling of exchange and other biophysical parameters (e.g., T2 relaxation, magnetization transfer), which are beyond the scope of the current study. Additional studies would also be necessary to uncover how molecular oscillations (L39-L57) translate into the MRI measurement rhythmicity.

We explored the potential contribution of hydration status as we had both the data and a mechanistic basis to do so (paragraph at L165). We also acknowledged that there could be some influence of external time givers (Zeitgebers) such as light, food, physical activity, which is why we used the term "diurnal" rather than "circadian" to describe the MRI-based oscillations throughout the study.

Reviewer #1 (Remarks to the Author):

This manuscript describes the results of a study that measured functional and structural MRI characteristics repeatedly (every three hours over 24 hours) in men with (n=16) and without (n=8) bipolar disorder (BD). Evidence for diurnal oscillations were observed in healthy individuals in white and gray matter mean diffusivity, white matter fractional anisotropy, and cerebral blood flow (CBF) and in gray and white matter quantitative relaxation time for meta-analytic analyses that were agnostic to acrophase; results held after controlling for significant body weight oscillations. Similar oscillations were observed regionally, particularly for CBF. BD showed similar oscillations for gray matter mean diffusivity and white matter fractional anisotropy, but not CBF or white matter mean diffusivity. BD showed larger phase variation in brain perfusion and the degree of deviation from the group mean was correlated with subjective sleep quality. Depending on the time of day, there were two-fold differences between BD and controls in CBF. Combining the samples revealed evidence for diurnal oscillations in surface area and gray matter volume not seen in controls or BD alone. The authors interpret their findings to have implications for population MRI studies and identify the need for higher power studies to examine, in particular, regional oscillations. They also discuss that desynchronization of CBF might be a biomarker of risk for relapse or a way to understand sleep disturbance in the disorder.

The study is novel and important and the manuscript is well written and clear. Although the sample sizes are small, particularly for the BD group, the within-subject design was powered well for the global measures. Only men were studied, which is acknowledged as a limitation for generalizability.

Comment: *The methods are appropriate to the questions posed. I was a bit unclear about whether, in the regional analyses, the global effects were somehow controlled or regressed out in order to identify regionally-specific rhythms? Also, were the regional measures weighted at all by their relative size (and presumably, reliability)?*

Response: We did not apply any adjustments to the regional data. The observation that the regional acrophases fit within the confidence intervals of the whole-brain acrophases (L193-L198) pointed to limited variation in regional oscillations.

Reliability of measurements was assessed at the whole brain level (paragraph at L120). We did not directly investigate the role of ROI size or include it in any statistical approaches. We agree that the factors involved in the detection of regional rhythms may be of interest, and we have added this to the discussion (L320-L321).

Comment: *The conclusions follow logically from the findings. My main critique is that [i] the authors did not discuss possible biological or environmental bases for the observed diurnal oscillations. [ii] What mechanisms might lead to the particular shape of the rhythms seen for, say, CBF in controls? Particularly for the more structural measures, what's going on in the brain that leads to different diffusivity at different times of the day? In addition, [iii] it might help researchers in the field for the authors to make some*

concrete recommendations about how to best “consider” these oscillations (especially the ones that are asynchronous across individuals) in cross-sectional designs given that most studies will not be able to scan participants 9 times in one day.

Response:

i. Our introductory comments above provide an overview on this point.

ii. In this study we focused on diurnal oscillations, which by corollary implies that the shape of variation is a 24-hr periodic function. Other rhythms, such as infradian (longer than 24 hr period) and ultradian (shorter than 24 hr period), may have also contributed to the variation of MRI measurements (L345), and these additional rhythms can be pursued in future investigations.

Evidence for CBF oscillations is not surprising given that CBF is tightly regulated to meet the brain's metabolic demands and that cell metabolism is an important part of the circadian machinery.

Mechanisms driving the oscillations in the more structural measurements are less clear, and this matter is briefly discussed above, in the introductory paragraph.

iii. There are several strategies that could be implemented to minimize diurnal confounders in cross-sectional designs: time-of-day matching, randomization, recording time-of-day, and modeling (L323-L325). We have also demonstrated that testing at a particular phase of the oscillatory differences may provide better case/control discriminability (Figure 4e). The implications of asynchronous oscillations (as in qT1) for cross-sectional studies are not immediately clear. The only apparent method to control for this would be temporal sampling of each subject. The generic strategies listed above may have little to no impact in cases like qT1.

Reviewer #2 (Remarks to the Author):

The authors present a very interesting study on a circadian oscillation in different MRI measures in the brain. Scanning protocol including sessions every 3h for 24h is a novel approach that was not introduced in previous neuroimaging research. The work will undoubtedly be of great importance in the field of chronobiology and neuroimaging. I only have a few remarks.

Comment: *The analysis was performed on a relatively small group. Have sample size calculations been performed?*

Response: Accurate power calculations could not have been performed for this study because there were no precedents that would provide an estimate of the effect size of oscillations and their patient-control differences. We acknowledged the apparent limitations of power seen in our regional results (L369-L371). Future circadian/diurnal MRI studies can use our results for sample size calculations.

Comment: *According to the General Procedure description, all subjects started scanning protocol at the same time (approx. 9 am). Did the authors consider the randomization of starting sessions across 24h?*

Response: This was considered, however, randomizing the start times could have added new confounding effects. For example, for subjects starting the scanning protocol in the evening or at night, the next day's data could be affected by disturbed sleep. For this initial study, we prioritized the detectability of oscillations over the exploration of variability introduced by factors such as start time.

Comment: *The authors declared to share processed data. Would you consider uploading raw (only anonymized) data to an open repository? This would be of great benefit and enable further analysis of this valuable data set.*

Response: At present, it is not possible to release the raw data. We are in discussions with our Privacy and Ethics offices about what kinds of unprocessed and/or minimally processed imaging data can be

shared with the scientific community.

Comment: *In the Discussion section, there is a lack of results description in relation to the literature, as well as a discussion of the results in the context of circadian changes in the physiology of the brain.*

Response: Our introductory comments to the reviewers address this point. Additionally, we have pointed to representative relevant literature in the introduction (L36-L71) and discussion (L304).

Reviewer #3 (Remarks to the Author):

In this article, the authors investigated with the help of repeated MRI acquisitions diurnal oscillations in a group of healthy adults and a smaller group of bipolar disorder patients.

The article is well-written and focuses on an important aspect which is broadly overlooked or neglected in many neuroimaging studies. Further, the authors used an advanced analysis method to detect possible oscillatory effects in various MRI-based measures. However, the study got abruptly stopped because of the COVID-19 pandemic.

Comment: *Although the study is well-designed and well-performed, this is a substantial limitation of the study, which propagates through all analyses and the validity of the conclusions.*

Response: Despite the premature termination of the project, we achieved the main objective of the project by demonstrating that diurnal oscillations are real and common among the MRI measurements.

Comment: *My main concern is the combination of the two groups. There are several issues with that procedure. Of course, it is unfortunate that the BDP group is so small, but it is still reasonable to expect at least some deviations.*

First of all, I strongly recommend not combining the two groups and then arguing whether significant results improved or not and using this as a proxy for whether BDP groups differed in this parameter or not. Given that there are potential differences, like in the CBF measure, indicates that there are potential differences between the groups, as predicted. The absence of clear effects in the BDP group for other parameters does not indicate that there are no effects. Lumping together the two groups on this weak evidence will primarily increase the chance of generating a statistical artefact rather than a valid result.

Response: In order to avoid the connotation that group differences were judged based on the combined analysis, we removed these implied comparisons (revised paragraph at L228 and L276-L277) or clearly explained the conclusions derived from them (L249). We have also made it clear that the main new finding arising from the combined analysis was that GMV was shown to have evidence of oscillations, which was not seen when the groups were analyzed separately.

Comment: *Secondly, the authors might also consider skipping the BDP group for this report and only reporting on the control group. This would make the study much more stringent and easier to interpret. I also encourage the author to continue their BDP study as planned. With the results provided now, it is difficult to draw any meaningful conclusions for this particular group.*

Response: The BPD data were useful to achieve our primary goal and conclude that MRI measurements oscillate with 24-hr periodicity. Our secondary goal was to test a BPD patient cohort for putative diurnal aberrations, which revealed an increased phase variability in CBF. We believe that these pilot BPD-related findings are also interesting and worth reporting.

Comment: *Thirdly, I'm missing an in-depth discussion on the causality of the observed effects. Why do the observed changes occur? Could it be that the observed effects in GM-MD and WM-MD are indirect effects of variations in CBF? I'm thinking here along the line of partial volume effects in the diffusion data*

caused by CBF (and perhaps related CBV) effects.

Response: Our introductory comments address the first question. Regarding the additional questions, the impact of CBF oscillations on WM-MD would be expected to be quite small, given the vasculature in white matter, and that we are measuring WM-MD on the TBSS skeleton, which is not adjacent to GM or ventricles. We also note that ASL is not a generally accepted method for measuring blood flow in white matter, and as such we do not report CBF values for the white matter skeleton. With respect to GM-MD and GM-CBF, the 5-hr difference in acrophase between the two metrics (16hr for GM-MD, 21hr for CBF, whole brain) argues against a direct influence of CBF on MD.

Comment: *Similarly, the introduction should also be revised to give a clearer picture of why these measures have been selected and what the expectations are.*

Response: We have added to the introduction to indicate the rationale behind our choice of metrics (L74-L75). Due to the lack of prior studies of this kind, it was not possible to set concrete expectations.

Other comments:

- Line 202 & 203: How were the results interrogated for evidence of oscillations? Only by eyeballing? And how was the similarity assessed?

Response: This text was intended as an overview, however, we have edited it to be more specific (L218-L219).

- Line 354 & line 377: There is conflicting information on how long the participants wear the actigraphs

Response: We have clarified by making the following change at L407: “wore a FitBit Flex™ for 4-5 days to achieve at least 3 full days of recordings”

- Line 361: Did the authors collect information on how much coffee, tea, or nicotine the participants consumed between scanning?

Response: Caffeine quantity was not measured and time of day was not restricted (now more clearly specified at L415-L416). Nicotine was not consumed during the experiment (L782; Table 1) “none consumed nicotine during the 24 hour scanning session”.

- Similarly, did the author collect data for blood pressure?

Response: No, we did not.

I'm looking forward to seeing a revised version of this manuscript

Karsten Specht

Reviewer #1 (Remarks to the Author):

The authors have thoroughly addressed my concerns.

Reviewer #2 (Remarks to the Author):

Thank you for responding to all my comments.

As authors pointed out, all the reviewers had comments regarding discussion of results. I still think that this section needs improvement. I understand the lack of additional biophysics metrics and thus lack of straight forward explanations. However, it would be of great benefit to the manuscript if the results were discussed in the context of physiology of the brain and its diurnal/circadian change. Also, authors cite previous paper regarding oscillation of MRI metrics (ref 8-17) but there is no discussion of current findings in relation to published results.

I would like to additionally add my support to Rev#3 comment about skipping the BDP group. Please consider again this suggestion and consider continuation of data acquisition for BDP group.

Reviewer #3 (Remarks to the Author):

The authors have adequately responded to my comments and concerns and revised the manuscript accordingly.

I have no further comments.

RESPONSE TO THE REVIEWERS' COMMENTS

Reviewer #1 (Remarks to the Author):

The authors have thoroughly addressed my concerns.

Reviewer #2 (Remarks to the Author):

Thank you for responding to all my comments.

Comment: *As authors pointed out, all the reviewers had comments regarding discussion of results. I still think that this section needs improvement. I understand the lack of additional biophysics metrics and thus lack of straightforward explanations. However, it would be of great benefit to the manuscript if the results were discussed in the context of physiology of the brain and its diurnal/circadian change. Also, authors cite previous paper regarding oscillation of MRI metrics (ref 8-17) but there is no discussion of current findings in relation to published results.*

Response: we have added the following paragraph to the Discussion (L343-352).

Text for Discussion:

“Uncovering the mechanisms for the observed diurnal oscillations, as well as previous time-of-day effects⁸⁻¹⁷, is part of a major and fundamental question: how are MRI measurements impacted by a variety of lower-level physiological and molecular brain processes. We made an initial step in this direction by determining that brain water content did not explain our results (Supplementary Table 3). The broad range of whole-brain acrophases across MRI metrics (Table 2) suggests other singular explanatory factors are implausible. Even data derived from a single pulse sequence (DTI) showed whole-brain acrophase variability from 13-18hr (GM-MD:16hr; WM-MD:13hr; WM-FA:18hr, Table 2). Multifaceted, multi-level approaches are necessary to reveal the mechanisms for MRI oscillations and to explain their individual, regional, and disease-related differences.”

Comment: *I would like to additionally add my support to Rev#3 comment about skipping the BDP group. Please consider again this suggestion and consider continuation of data acquisition for BDP group.*

Response: We strongly believe that adding the BPD group had more benefits than shortcomings. Due to the lengthy COVID lockdown, we were able to collect only part of the planned sample, and every individual, whether a control or patient, was of great value. In our opinion, adding the BPD group, and analysing it with controls, was a useful heuristic step to demonstrate that the majority of the MRI measurements oscillate with 24-hr periodicity. If any doubts about the contribution of the BPD group arise to the readership, they can refer to the controls-only results which are fully provided in the paper.

In addition, as stated in our previous response, one of our goals was to test a BPD patient cohort for putative diurnal aberrations. In spite of the small sample size, we found significant evidence of increased CBF phase variability in the patients. We believe it is important to present these data and offer some new directions for future BPD studies.

With respect to continuation of the study, a completely new sample of patients and matched controls will be required. Given the significant gap (>3yrs) from when we were last able to acquire data, scanner upgrades, and possible contribution of various other confounding factors, it would be unwise to simply combine datasets. In our future studies, we plan to use the current datasets for further

optimization of our imaging protocols which will help in teasing out mechanisms for the observed oscillations in both patients and controls.

We would also like to note that our previous response to the comment about the *pros* and *cons* of the BPD group, as well as changes in the text of the manuscript, satisfied the 3rd reviewer (who originally raised this issue).

Reviewer #3 (Remarks to the Author):

The authors have adequately responded to my comments and concerns and revised the manuscript accordingly.

I have no further comments.